# Species Diversity and Ecological Habitat of *Absidia* (Cunninghamellaceae, Mucorales) with Emphasis on Five New Species from Forest and Grassland Soil in China

**DOI:** 10.3390/jof8050471

**Published:** 2022-04-30

**Authors:** Heng Zhao, Yong Nie, Tong-Kai Zong, Yu-Jie Wang, Mu Wang, Yu-Cheng Dai, Xiao-Yong Liu

**Affiliations:** 1Institute of Microbiology, School of Ecology and Nature Conservation, Beijing Forestry University, Beijing 100083, China; zhaoheng21@bjfu.edu.cn; 2College of Life Sciences, Shandong Normal University, Jinan 250014, China; 3School of Civil Engineering and Architecture, Anhui University of Technology, Ma’anshan 243002, China; lyly19851207@126.com; 4Key Laboratory for Forest Resources Conservation and Utilization in the Southwest Mountains of China, Ministry of Education, Southwest Forestry University, Kunming 650224, China; zongfungi@163.com; 5College of Plant Science, Tibet Agricultural and Animal Husbandry University, Linzhi 860000, China; 18503217053@163.com (Y.-J.W.); wangmutb@163.com (M.W.)

**Keywords:** fungal diversity, geographical distribution, early diverging fungi, mucoromycota

## Abstract

Although species of *Absidia* are known to be ubiquitous in soil, animal dung, and insect and plant debris, the species diversity of the genus and their ecological habitats have not been sufficiently investigated. In this study, we describe five new species of *Absidia* from forest and grassland soils in southwestern China, with support provided by phylogenetic, morphological, and physiological evidence. The species diversity and ecological habitat of *Absidia* are summarized. Currently, 22 species are recorded in China, which mainly occur in soil, especially in tropical and subtropical forests and mountains. An updated key to the species of *Absidia* in China is also provided herein. This is the first overview of the *Absidia* ecological habitat.

## 1. Introduction

The genus *Absidia* Tiegh., typified by *A. reflexa* Tiegh., was described nearly 150 years ago [1], belonging to Cunninghamellaceae, Mucorales, Mucoromycetes, Mucoromycota (http://www.indexfungorum.org/, accessed on 1 March 2022). However, it was initially placed in the family Absidiaceae [2]. With the development of molecular biology, it was grouped with *Chlamydoabsidia* Hesselt. and J.J. Ellis, *Cunninghamella* Matr., *Gongronella* Ribaldi, and *Halteromyces* Shipton and Schipper and *Hesseltinella* H.P. Upadhyay; this group was nominated as the family Cunninghamellaceae [3,4,5,6,7]. Nine other genera were thought to be allied with *Absidia* on the basis of morphological similarities, including *Rhizopus* Ehrenb. 1821, *Phycomyces* Kunze 1823, *Tieghemella* Berl. and De Toni 1888, *Mycocladus* Beauverie 1900, *Lichtheimia* Vuill. 1903, *Proabsidia* Vuill. 1903, *Pseudoabsidia* Bainier 1903, *Protoabsidia* Naumov 1935, and *Gongronella* Ribaldi 1952 [8]. As research progressed, *Absidia* s.l. has been well divided into three genera, i.e., *Lichtheimia* (thermotolerant, optimum growth temperature 37–45 °C), *Absidia* s.s. (mesophilic, optimum growth temperature 25–34 °C), and *Lentamyces* Kerst. Hoffm. and K. Voigt (parasitic on mucoralean fungi, optimum growth temperature 14–25 °C) [9,10,11]. Currently, species of *Absidia* are characterized by (1) sporangiophores single, in pairs or in groups on stolons, (2) rhizoids at both ends of stolons and never opposite the sporangiophores, (3) sporangia deliquescent-walled and apophysate, (4) columellae bearing one to several projections, (5) zygospores enclosed by appendages, and (6) optimum growth temperatures from 25 °C to 34 °C [1,9,10,11,12,13,14,15].

So far, a total of 46 species of *Absidia* have been described worldwide, and they are ubiquitous in soil, dung, insect, leaf litter, food, air, etc. (Figure 1; [16]). Among them, 30 species were initially collected from soil, including forest and rhizosphere soil, suggesting that *Absidia* species are mainly isolated from soil [17,18]. However, a few species are endemic to animal dung and insect remains, such as *A. psychrophilia* Hesselt. and J.J. Ellis from mycangia of ambrosia beetles [8,19], and *A. stercoraria* Hyang B. Lee et al. from rat dung [20].

*Absidia glauca* Hagem and *A. repens* Tiegh. are considered as cosmopolitan species, reported in all continents except Antarctica [21,22,23]. *Absidia heterospora* Y. Ling was reported in China, New Zealand, and France [23,24]; *A. idahoensis* Hesselt. et al. and *A. macrospora* Váňová were reported in China, Czechia, and the USA [25,26,27]. Since 2018, 22 endemic species have been described from Korea, China, Thailand, Australia, USA, Mexico, and Brazil [14,15,16,28,29,30,31,32,33,34,35,36]. Type strains were collected from 17 countries, and the two most investigated countries are China and Brazil, with nine and eight type strains, respectively. The studies on *Absidia* in other countries and regions are obviously insufficient (Table 1).

Therefore, there are deficiencies in the studies on species distribution and ecological habitat of *Absidia* [8,13,14,15,16,18,20,27]. In this paper, we propose five new species from forest and grassland soil in Sichuan, Tibet, and Yunnan in southwestern China. A key to *Absidia* species in China is consequently updated. Along with the taxonomical study, we also conduct a preliminary investigation on the species distribution and ecological habitat of *Absidia*.

## 2. Materials and Methods

### 2.1. Isolation and Strains

Strains were isolated from forest and grassland soil samples collected in September 2021, in Sichuan, Tibet, and Yunnan in southwestern China. An aliquot of soil samples (1 g) was evenly spread on 15 cm petri dishes containing potato dextrose agar medium (200 g potato, 20 g dextrose, 20 g agar, and 1000 mL distilled water) with streptomycin sulfate and ampicillin 100 mg/mL each, and then cultivated at 20 °C and 25 °C. According to morphological characteristics of *Absidia*, potential strains were picked out and purified. The purified living cultures (Table 2) were deposited in the China General Microbiological Culture Collection Center, Beijing, China (CGMCC) and the Shandong Normal University (XY), and dry cultures (Table 2) were deposited in the Herbarium Mycologicum Academiae Sinicae, Beijing, China (HMAS).

### 2.2. Morphology and Maximum Growth Temperature

Pure cultures were incubated with malt extract agar medium (MEA: 30 g malt extract, 3 g peptone, 20 g agar, and 1000 mL distilled water; [37]). For morphological studies, two plates of each strain were cultured at 20 °C and 27 °C, respectively, and then examined under a stereomicroscope (SMZ1500, Nikon, Tokyo, Japan) and a light microscope (Axio Imager A2, Carl Zeiss, Oberkochen, Germany). Maximum growth temperature tests followed the methods in our previous studies [14,15,16,38,39,40,41]. For maximum growth temperature tests, three plates were incubated at 20 °C for 2 d, then incubation temperature was increased by a gradient of 1 °C until the colonies stopped growing. For morphological features, the minimum and maximum sizes based on a statistic of more than 50 measurements were adopted [16]. All cultures were triplicated.

### 2.3. Molecular Phylogeny

DNA extraction, PCR amplification, and sequencing and phylogenetic analyses followed the methods in our previous studies [14,16,42,43,44]. In brief, total DNAs were extracted using a kit (GO-GPLF-400, GeneOnBio Corporation, Changchun, China) based on the instruction manual. Entire ITS and partial LSU rDNA sequences were amplified with primer pairs NS5M (5′-GGC TTA ATT TGA CTC AAC ACG G-3′) and LR5M (5′-GCT ATC CTG AGG GAA ACT TCG-3′; [14,16]). PCR procedures were as follows: an initial denaturation at 95 °C for 5 min, 30 cycles at 95 °C for 60 s, 55 °C for 45 s, and 72 °C for 60 s, and a final extra extension at 72 °C for 10 min. Sanger sequencing of PCR products were carried out by a company (SinoGenoMax, http://www.sinogenomax.com, accessed on 1 March 2022) with primers ITS1, ITS4, ITS5, and LR5M [14,16,45].

Phylogenetic analyses were conducted with maximum likelihood (ML), maximum parsimony (MP) and Bayesian inference (BI) algorithms [39,41] using RAxML (version 8. 1.12; [46]), PAUP (version 4.0b10; [47]) and MrBayes (version 3.2.7a; [48]), respectively. Maximum Likelihood analysis was performed using the GTRGAMMA substitution model with 1000 bootstrap replications. Maximum parsimony analysis was conducted with 1000 bootstrap replications using the heuristic search option with bisection and reconnection. Sequences were randomly added and max-trees were set to 5000. For BI analysis, eight cold Markov chains were run simultaneously for two million generations with the GTR + I + G model, sampling every 1000 generations and with the first 25% sampled tree being removed as burn-in. Obtained sequences and aligned dataset were deposited at GenBank (Table 2) and Appendix A, respectively. Additionally, top hits of the BLAST search for ITS sequences are provided in the Appendix A.

## 3. Results

### 3.1. Phylogenetic Analyses

The ITS and LSU rDNA dataset for 62 strains, representing 45 species of *Absidia* and two species of *Cunninghamella*, contains 1642 characters, of which 501, 877, and 264 are constant, parsimony-informative, and parsimony-uninformative, respectively. Maximum parsimony (MP) analyses yielded two equal trees (tree length 6669, consistency index 0.3330, homoplasy index 0.6670, retention index 0.5669, and rescaled consistency index 0.1888). The most optimal model of Bayesian inference (BI) was GTR + I + G, and the average standard deviation of split frequencies was 0.07357. Topology of the ML tree was chosen to represent the phylogenetic relationship (Figure 2). In the phylogenetic tree, the five new species described herein are fully supported with *Cunninghamella elegans* Lendn. and *Cunninghamella blakesleeana* Lendn. as outgroup.

### 3.2. Taxonomy

In this paper, five new species of *Absidia* are proposed from southwestern China, i.e., Sichuan, Tibet, and Yunnan. Besides the ITS and LSU rDNA sequences provided above, all the new species are illustrated along with morphological characteristics and the maximum growth temperature; a physiological trait is also presented.

#### 3.2.1. ***Absidia abundans*** H. Zhao, Y.C. Dai and X.Y. Liu, sp. nov.

*Fungal Names*: FN570973. Figure 3.

*Etymology*: *abundans* (Lat.) refers to the species with abundant swellings below the sporangia.

*Holotype*: HMAS 351932.

*Description*: Colonies on MEA at 27 °C for 7 days, growing moderately fast, attaining 65 mm in diameter, initially white, soon becoming gray to brown, and irregularly and concentrically zonate with ring at reverse. Hyphae are branched, hyaline at first, brownish when mature, aseptate when juvenile, septate with age, and 3.0–15.5 µm in diameter. Stolons branched, hyaline, and smooth. Rhizoids are root-like, often unbranched, and well-developed. Sporangiophores arise from stolons, always erect, unbranched, or simple if branched, monopodial, or sympodial, hyaline, swellings usually present below sporangia, oval to pyriform, hyaline, often with a septum 12.0–17.0 µm below apophyses, 35.0–170.0 µm long, and 2.0–3.5 µm wide. Sporangia are oval to subglobose, hyaline when young, dark green when old, smooth, deliquescent-walled, multi-spored, 8.0–16.5 µm long, and 8.5–16.0 µm wide. Apophyses are distinct, hyaline or subhyaline, 2.5–5.5 µm high, gradually widened upwards, 2.5–4.5 µm wide at the base, and 4.5–8.0 µm wide at the top. Collars absent or present. Columellae are subglobose or oval, hyaline, subhyaline or light brown, 4.5–10.0 µm long, and 3.5–8.0 µm wide. Projections are single if present, subhyaline, and 1.0–2.5 µm long. Sporangiospores are cylindrical, oval or subglobose, light green, smooth, 2.5–3.5 µm long, and 2.0–3.5 µm wide. Chlamydospores are absent. Zygospores are not observed.

*Maximum growth temperature*: 31 °C.

*Materials examined*: China. Yunnan, Qujing, from forest soil sample, September 2021, Heng Zhao (holotype HMAS 351932, living ex-holotype culture CGMCC 3.16255, and living cultures XY09265 and XY09274).

#### 3.2.2. ***Absidia lobata*** H. Zhao, Y.C. Dai and X.Y. Liu, sp. nov.

*Fungal Names*: FN570974. Figure 4.

*Etymology*: *lobata* (Lat.) refers to the species with a broadly lobed edge of colonies.

*Holotype*: HMAS 351933.

*Description*: Colonies on MEA at 20 °C for 7 days, growing moderately fast, attaining 70 mm in diameter, zonate, broadly lobed at edge, white at first, gradually becoming light to dark brown and greenish, and irregular at reverse. Hyphae are branched, hyaline at first, greenish brown when mature, aseptate when juvenile, septate with age, and 4.0–21.0 µm in diameter. Stolons are branched, hyaline, brownish or light greenish, smooth, and septate. Rhizoids are sometimes unbranched and rootlike when branched. Sporangiophores arising from stolons, in pairs, unbranched, erect or slightly bent, hyaline or light brown, sometimes with a septum 13.0–22.5 µm below apophyses, 50.0–360.0 µm long, and 3.0–7.0 µm wide. Sporangia are pyriform, colorless when young, light to dark brown when old, smooth, deliquescent-walled, multi-spored, 22.0–43.5 µm long, and 18.5–31.0 µm wide. Apophyses are always distinct, hyaline to light brown, 4.0–9.0 µm high, gradually widened upwards, 4.5–8.5 µm wide at the base, and 10.0–16.0 µm wide at the top. Collars are always absent. Columellae are subglobose to depressed globose, rarely irregular, subhyaline or light brown, rough, 12.0–26.5 µm long, and 11.0–25.0 µm wide. Projections are always present, single, hyaline or subhyaline, and 2.0–6.0 µm long. Sporangiospores are mostly globose, occasionally subglobose, light green, smooth, and 2.5–3.0 µm in diameter. Chlamydospores are absent. Zygospores are not observed.

*Maximum growth temperature*: 26 °C.

*Materials examined*: China, Yunnan, Lijiang, 27°31′23″ N, 100°44′32″ E, altitude: 3153 m, from rhizosphere soil of *Pinus yunnanensis* Franch., September 2021, Heng Zhao (holotype HMAS 351933, living ex-holotype culture CGMCC 3.16256, and living culture XY08832-1).

#### 3.2.3. ***Absidia radiata*** H. Zhao, Y.C. Dai and X.Y. Liu, sp. nov.

*Fungal Names*: FN570975. Figure 5.

*Etymology*: *radiata* (Lat.) refers to the species with a radiate shape of colonies.

*Holotype*: HMAS 351934.

*Description*: Colonies on MEA at 27 °C for 7 days, growing moderately fast, attaining 65 mm in diameter, white at first, gradually becoming light to dark brown, with adjoining satellite colonies at the edge at reverse. Hyphae are branched, hyaline when young, light brownish when old, aseptate when juvenile, septate with age, and 3.0–16.0 µm in diameter. Stolons are branched, hyaline to light brown, and smooth. Rhizoids are rootlike, unbranched, and well-developed. Sporangiophores arise from stolons, 1–5 in whorls, erect or slightly bent, unbranched, hyaline to light brown, sometimes with a septum 16.5–24.5 µm below apophyses, 45.0–273.0 µm long, and 3.0–5.0 µm wide. Sporangia are pyriform or subglobose, hyaline when young, brown when old, smooth, deliquescent-walled, multis-pored, 17.5–33.5 µm long, and 18.5–30.0 µm wide. Apophyses are distinct, hyaline to light brown, 5.5–12.5 µm high, with a turning point at the top of sporangiospores, then gradually widened upwards, 4.0–5.5 µm wide at the base, and 9.0–20.0 µm wide at the top. Collars absent. Columellae are mostly oval, depressed globose, occasionally subglobose to globose, subhyaline or hyaline, 13.5–22.5 µm long, and 14.0–24.0 µm wide. Projections are present or absent, single if present, subhyaline, and 2.0–4.5 µm long. Sporangiospores are oval, subhyaline, smooth, 3.0–5.0 µm long and 2.0–3.5 µm wide. Chlamydospores are absent. Zygospores are not observed.

*Maximum growth temperature*: 32 °C.

*Materials examined*: China, Yunnan, Yuxi, from forest soil sample, September 2021, Heng Zhao (holotype HMAS 351934, living ex-holotype culture CGMCC 3.16257, and living culture XY09330-1).

#### 3.2.4. ***Absidia sichuanensis*** H. Zhao, Y.C. Dai and X.Y. Liu, sp. nov.

*Fungal Names*: FN570977. Figure 6.

*Etymology*: *sichuanensis* (Lat.) refers to the species found in Sichuan Province, southwest China.

*Holotype*: HMAS 351935.

*Description*: Colonies on MEA at 20 °C for 7 days, growing moderately fast, attaining 75 mm in diameter, white at first, soon becoming light brown, irregularly concentrically zonate at reverse. Hyphae are branched, hyaline at first, greenish brown when mature, aseptate when juvenile, septate with age, 5.5–15.5 µm in diameter. Stolons are branched, hyaline to brownish, smooth. Rhizoids are sometimes unbranched, fingerlike or rootlike when branched, and well-developed. Sporangiophores arise from stolons, 1–5 in whorls, unbranched, erect or slightly bent, hyaline, sometimes with a septum 14.0–16.5 µm below apophyses, 30.0–220.0 µm long, and 3.0–5.0 µm wide. Sporangia are pyriform, greenish when old, smooth, deliquescent-walled, multi-spored, 18.0–23.0 µm long and 17.0–21.5 µm wide. Apophyses distinct, hyaline or subhyaline, 2.0–6.5 µm high, gradually widened upwards but with a turning point at the top of sporangiophores, 3.0–5.0 µm wide at the base, and 5.5–12.0 µm wide at the top. Collars are usually absent and rarely present. Columellae are mostly subglobose, occasionally globose, hyaline to greenish, 7.5–13.0 µm long, and 8.0–15.5 µm wide. Projections present or absent, single if present, hyaline or subhyaline, 4.0–6.0 µm long. Sporangiospores are cylindrical, subhyaline, smooth, 3.0–4.5 µm long and 2.0–2.5 µm wide. Chlamydospores are absent. Zygospores are not observed.

*Maximum growth temperature*: 28 °C.

*Materials examined*: China. Sichuan, Ngawa, 30°42’18” N, 101°22’17” E, altitude: 3727 m, from grassland soil sample, September 2021, Heng Zhao (holotype HMAS 351935, living ex-holotype culture CGMCC 3.16258), from rhizosphere soil of *Picea asperata* Mast., Heng Zhao (living culture XY09119). Tibet, Bome, from rhizosphere soil of *Pinus yunnanensis*, September 2021, Heng Zhao (living culture XY09633).

#### 3.2.5. ***Absidia yunnanensis*** H. Zhao, Y.C. Dai and X.Y. Liu, sp. nov.

*Fungal Names*: FN570978. Figure 7.

*Etymology*: *yunnanensis* (Lat.) refers to the species found in Yunnan Province, southwest China.

*Holotype*: HMAS 351936.

*Description*: Colonies on MEA at 27 °C for 7 days, growing moderately fast, attaining 65 mm in diameter, initially white, soon becoming light brown, and irregularly and concentrically zonate with ring at reverse. Hyphae are branched, hyaline at first, light brown when mature, aseptate when juvenile, septate with age, and 4.0–15.5 µm in diameter. Stolons are branched, hyaline, light brown or green, smooth. Rhizoids rootlike, often unbranched, and well-developed. Sporangiophores arise from rhizoids, 2–5 in whorls, erect or slightly bent, usually unbranched, rarely monopodially branched, hyaline, light brown or green, swellings usually present below sporangia, usually oval, rarely irregular, hyaline, often with a septum 12.0–16.5 µm below apophyses, 29.0–159.0 µm long, and 2.0–5.0 µm wide. Sporangia are pyriform to subglobose, hyaline when young, light green when old, smooth, deliquescent-walled, multi-spored, 16.0–27.5 µm long, and 16.5–27.0 µm wide. Apophyses are distinct, hyaline or subhyaline, 3.0–6.5 µm high, gradually widened upwards, 3.0–7.5 µm wide at the base, and 5.5–14.0 µm wide at the top. Collars are absent or present. Columellae are oval, depressed globose, or occasionally globose, hyaline, subhyaline or light green, 6.5–18.5 µm long, and 7.5–22.0 µm wide. Projections are present or absent, single if present, subhyaline, and 1.5–3.5 µm long. Sporangiospores are cylindrical, light green, smooth, 3.5–5.0 µm long, and 2.0–4.0 µm wide. Chlamydospores are absent. Zygospores are not observed.

*Maximum growth temperature:* 32 °C.

*Material examined:* China. Yunnan, Yuxi, from forest soil sample, September 2021, Heng Zhao (holotype HMAS 351936, living ex-holotype culture CGMCC 3.16259). Chuxiong, 25°13’52” N, 101°18’24” E, altitude: 2060 m, from forest soil sample, September 2021, Heng Zhao (living culture XY09528).

### 3.3. Key to the Species of Absidia in China

Together with the five new species proposed in this study, a total of 51 species of *Absidia* have been described worldwide. Among these, 22 species are distributed in China. Consequently, we provide a key to the Chinese species of *Absidia*. Characteristics adopted in the key include maximum growth temperatures, hyphae, rhizoids, sporangiophores, sporangia, collars, columellae, projections, and sporangiospores.
1. Maximum growth temperature ≤ 28 °C21. Maximum growth temperature > 28 °C62. Sporangiospores subglobose to globose32. Sporangiospores cylindrical43. Maximum growth temperature 26 °C; sporangiospores subglobose to globose*A. lobata*3. Maximum growth temperature 28 °C; sporangiospores globose*A. globospora*4. Maximum growth temperature 24 °C*A. frigida*4. Maximum growth temperature 28 °C55. Sporangiophores > 5 in whorls*A. psychrophilia*5. Sporangiophores ≤ 5 in whorls*A. sichuanensis*6. Sporangiospores two or more types76. Sporangiospores one type127. Sporangiospores sometimes irregular in shape107. Sporangiospores never irregular in shape*A. globospora*8. Sporangia elliptical or elongate*A. repens*8. Sporangia globose to pyriform99. Hyphae without swellings; sporangiophores monopodial or verticillate*A. idahoensis*9. Hyphae with swellings; sporangiophores sometimes unbranched*A. turgida*10. Columellae with projections*A. abundans*10. Columellae without projections1111. Sporangiospores globose, 3.8–7.7 µm in diameter, or cylindrical to oval*A. heterospora*11. Sporangiospores globose, 2.5–3.5 µm in diameter, or cylindrical*A. gemella*12. Sporangiospores globose*A. glauca*12. Sporangiospores cylindrical, oval or ellipsoid1313. Collars absent1413. Collars present1614. Sporangiospores cylindrical*A. longissima*14. Sporangiospores oval or ellipsoid1515. Sporangiospores oval to ellipsoid; sporangiophores 2–6 in whorls with swellings*A. ovalispora*15. Sporangiospores oval; sporangiophores 2–5 in whorls without swellings*A. radiata*16. Sporangiophores neither in pairs nor in whorls*A. panacisoli*16. Sporangiophores in pairs and in whorls1717. Sporangiophores 7–11 in whorls1817. Sporangiophores ≤ 6 in whorls2018. Rhizoids aseptate*A. spinosa*18. Rhizoids septate1919. Projections < 5 µm long, tapering at top*A. zonata*19. Projections > 5 µm long, rounded at top*A. pseudocylindrospora*20. Maximum growth temperature > 34 °C*A. cylindrospora*20. Maximum growth temperature ≤ 34 °C2121. Sporangiophores always swollen*A. yunnanensis*21. Sporangiophores never swollen*A. medulla*

### 3.4. Species Distribution and Ecological Habitat in China

A total of 22 species of *Absidia* were recorded in China (Table 3), including the five new species proposed in this study [14,15,16,27,49,50,51]. All species were found in soil, such as forest soil and rhizospheric soil, whereas other habitats, including leaf litter, dung, insect remains, and plant leaves, recorded one to several species. In terms of geographical distribution, Yunnan, Xinjiang, and Taiwan top the list with twelve, six, and five records, respectively.

## 4. Discussion

In this study, five new species are proposed in the genus of *Absidia* being supported with molecular sequences and morphological and physiological features. Phylogenetically, *Absidia abundans* is closely related to *A. panacisoli* T. Yuan Zhang et al. based on the ITS and LSU rDNA sequences (Figure 2). However, *A. panacisoli* is distinguished from *A. abundans* by a higher maximum growth temperature (33 °C vs. 31 °C), shape of sporangia (spherical or subpyriform vs. oval to subglobose), sporangiospores (short cylindrical vs. cylindrical, oval or subglobose), and azygospores (present vs. absent; [16]). Moreover, swellings below sporangia are always observed in *A. abundans*, while absent in *A. panacisoli* [16].

*Absidia lobata* is closely related to *A. glauca* and *A. globospora* T.K. Zong and X.Y. Liu (Figure 2), while distinguished by a lower maximum growth temperature (26 ℃ in *A. lobata,* 29 °C in *A. glauca*, 37 °C in *A. globospora*) [15,52]. Besides, sporangia are globose in *A. globospora*, while pyriform in *A. lobata* and *A. glauca* [15,52]. Moreover, zygospores and chlamydospores are produced in *A. glauca*, but not in *A. lobata* and *A. globospora* [15,52].

*Absidia radiata* is related to *A. yunnanensis* with full support (Figure 3). However, *A. yunnanensis* differs from *A. radiata* by colonies (light brown vs. dark brown), shape of sporangiospores (cylindrical vs. oval), and swellings below sporangia (present vs. absent).

*Absidia sichuanensis* is most closely related to *A. frigida* H. Zhao et al. and *A. psychrophilia* (Figure 4), but differs by a higher maximum growth temperature (28 °C in *A. sichuanensis*, 24 °C in *A. frigida*, and 25 °C in *A. psychrophilia*), sporangia (17.0–21.5 µm wide in *A. sichuanensis*, 12.5–32.0 µm wide in *A. frigida*, and 20.0–50.0 µm wide in *A. psychrophilia*), columellae (8.0–15.5 µm wide in *A. sichuanensis,* 15.5–18.0 µm wide in *A. frigida*, and 6.5–30.0 µm wide in *A. psychrophilia*), and number of whorls (1–5 in *A. sichuanensis*, 1–4 in *A. frigida*, and 1–8 in *A. psychrophilia* [8,16]. In addition, collars are always absent in *A. sichuanensis* and *A. frigida* but present in *A. psychrophilia* [8,16]. Moreover, a morphological feature table including six species of *Absidia* without DNA sequences is listed for distinguishing between the five new species proposed in this study (Table 4).

Species of *Absidia* synthesized important metabolites, such as α-galactosidase, laccase, chitosan, and fatty acids [57,58,59,60], and our results provide a basis for their applications. In addition to the five new species proposed in this paper, 51 species of *Absidia* have previously been described from all around the world. A total of 22 species are recorded in China (https://nmdc.cn/fungarium/fungi/chinadirectories, accessed on 1 March 2022), accounting for 43% of worldwide species of the genus *Absidia* [14,15,16,18,27,39,40,41,42,43,44,45,46,47,48,49,50,51]. This result suggests that *Absidia*, a genus of Mucoromycota, is diverse, needing in-depth investigations to discover and describe more potential new species, as in Ascomycota and Basidiomycota [13,61,62,63,64].

The species of *Absidia* are ubiquitous in soil, dung, and decaying plants, as well as insect remains [1,8,15,16,51]. Most species, including the five new species described herein, are reported from soil and, hence, soil is their main habitat. Some species may be associated with plants (Table 3). For example, *A. lobata* and *A. panacisoli* were described in rhizosphere soil with *Pinus yunnanensis* and *Panax notoginseng* (Burkill) F. H. Chen ex C. Chow and W. G. Huang, respectively [16].

In China, most species of *Absidia* are recorded in Yunnan, Xinjiang, and Taiwan (Table 3), located in tropical, subtropical, and temperate zones. At the same time, a number of *Absidia* are described from Brazil and Thailand, which have a similar climate [17,28,29,30,33,34]. However, species of *Absidia* are rarely adapted to high temperatures, so that strains in tropical areas are usually described from forest soil or mountains [16,17,28,29,33,34]. Consequently, species diversity of *Absidia* in tropical and subtropical forest soil needs to be further explored. 

## Figures and Tables

**Figure 1 jof-08-00471-f001:**
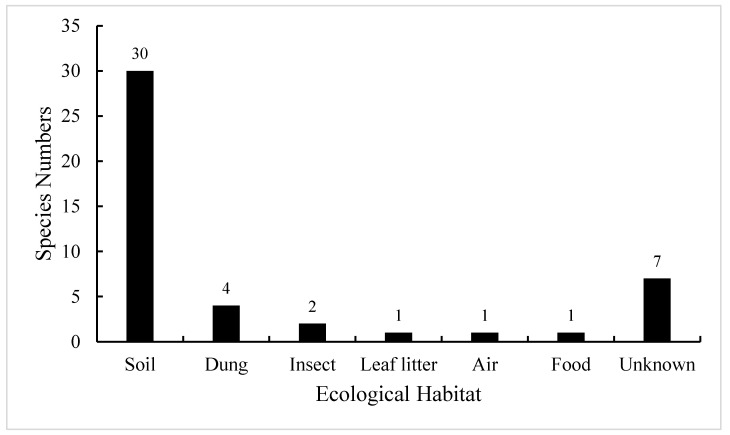
Number of *Absidia* species in different ecological habitats when they were originally described. These data are from the Index Fungorum (http://www.indexfungorum.org/, accessed on 1 March 2022) and [16].

**Figure 2 jof-08-00471-f002:**
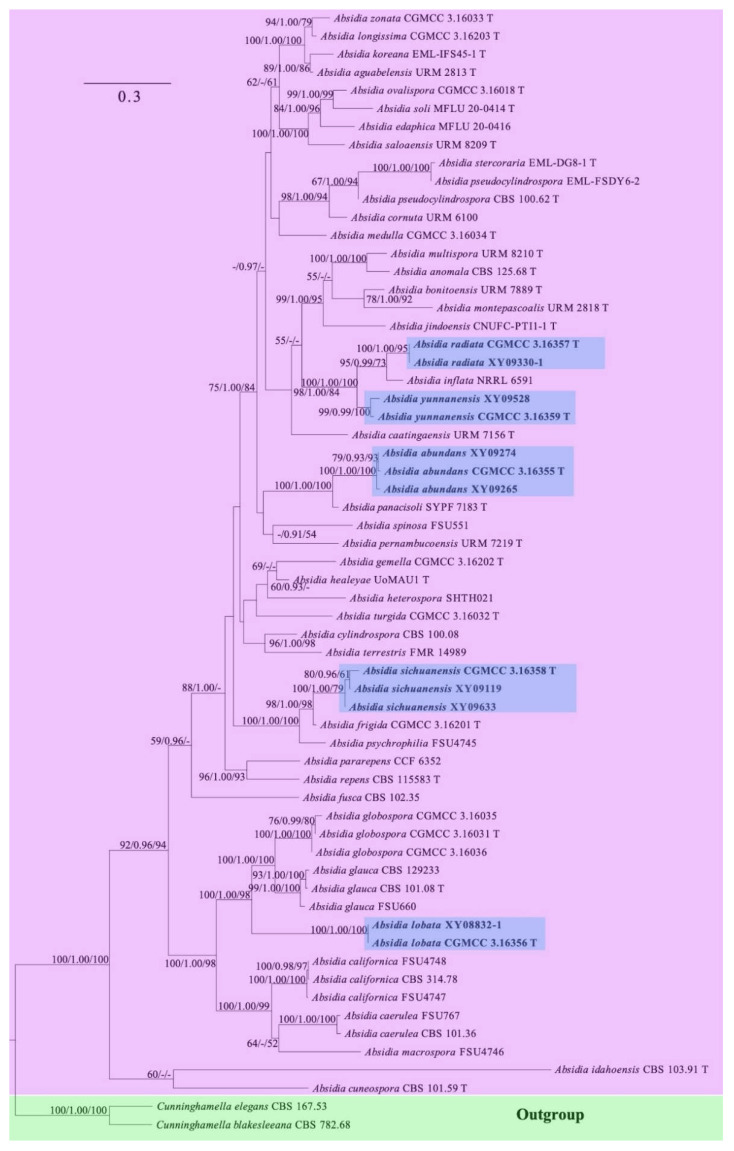
Maximum likelihood (ML) phylogenetic tree of *Absidia* based on ITS and LSU rDNA sequences, with *Cunninghamella elegans* and *C. blakesleeana* as outgroup shaded in blue. Five new species are in shade and bold. Maximum likelihood bootstrap values (≥50%)/Bayesian inference (BI) posterior probabilities (≥0.9)/maximum parsimony (MP) bootstrap values (≥50%) of each clade are indicated along branches. “T” after strain number represents type. A scale bar in the upper left indicates substitutions per site.

**Figure 3 jof-08-00471-f003:**
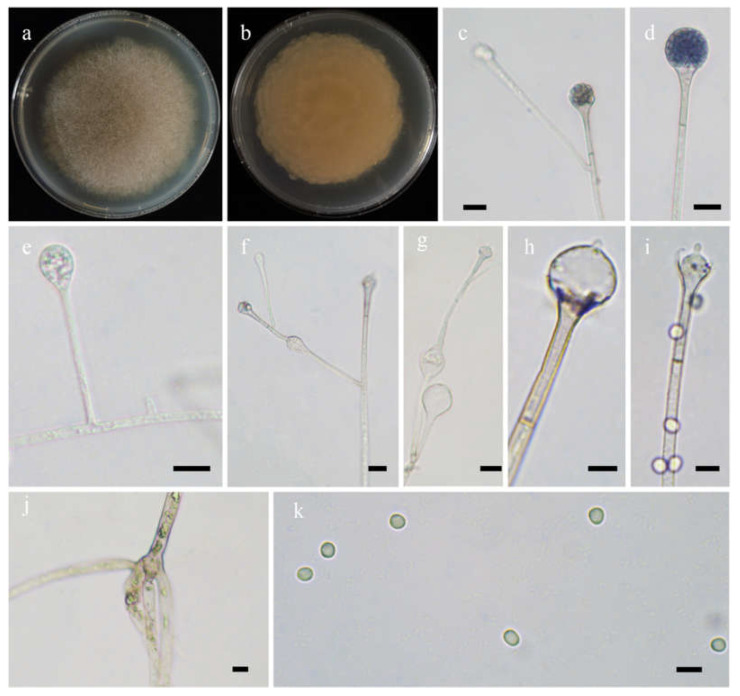
Morphologies of *Absidia abundans* ex-holotype CGMCC 3.16255. (**a**,**b**). Colonies on MEA ((**a**), obverse, (**b**), reverse); (**c**–**e**), sporangia; (**f**), sympodially branched sporangiophores; (**g**), swellings below sporangia; (**h**), columellae with projections; (**i**), columellae with projections and collars; (**j**), rhizoids; (**k**), sporangiospores—scale bars: (**c**–**g**), (**j**), 10 μm, (**h**,**i**,**k**), 5 μm.

**Figure 4 jof-08-00471-f004:**
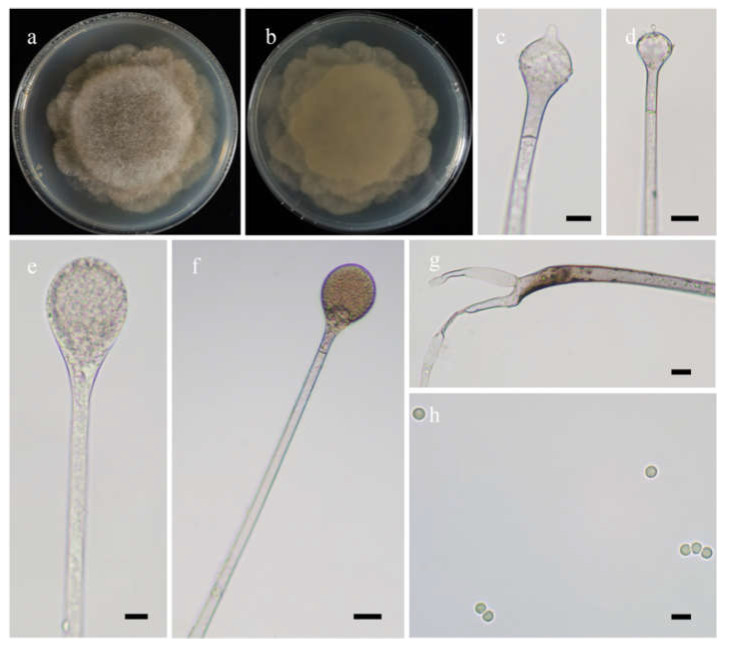
Morphologies of *Absidia lobata* ex-holotype CGMCC 3.16256. (**a**,**b**). Colonies on MEA ((**a**), obverse, (**b**), reverse); (**c**,**d**), columellae with projections; (**e**,**f**)c sporangia; (**g**), rhizoids; (**h**), sporangiospores—scale bars: (**c**–**e**), 10 μm, (**f**,**g**), 20 μm, (**h**), 5 μm.

**Figure 5 jof-08-00471-f005:**
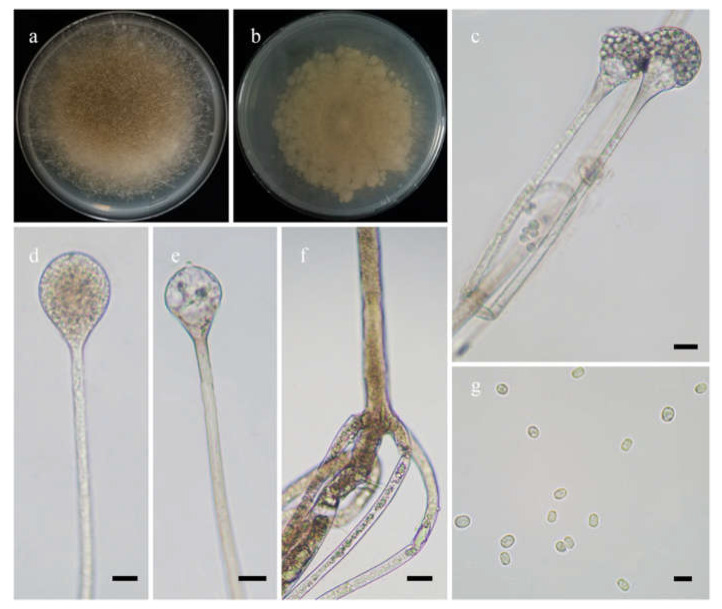
Morphologies of *Absidia radiata* ex-holotype CGMCC 3.16257. (**a**,**b**). Colonies on MEA ((**a**), obverse, (**b**), reverse); (**c**,**d**), sporangia; (**e**), columellae with projections; (**f**), rhizoids; (**g**), sporangiospores—scale bars: (**c**–**e**), 10 μm, (**f**), 20 μm, (**g**), 5 μm.

**Figure 6 jof-08-00471-f006:**
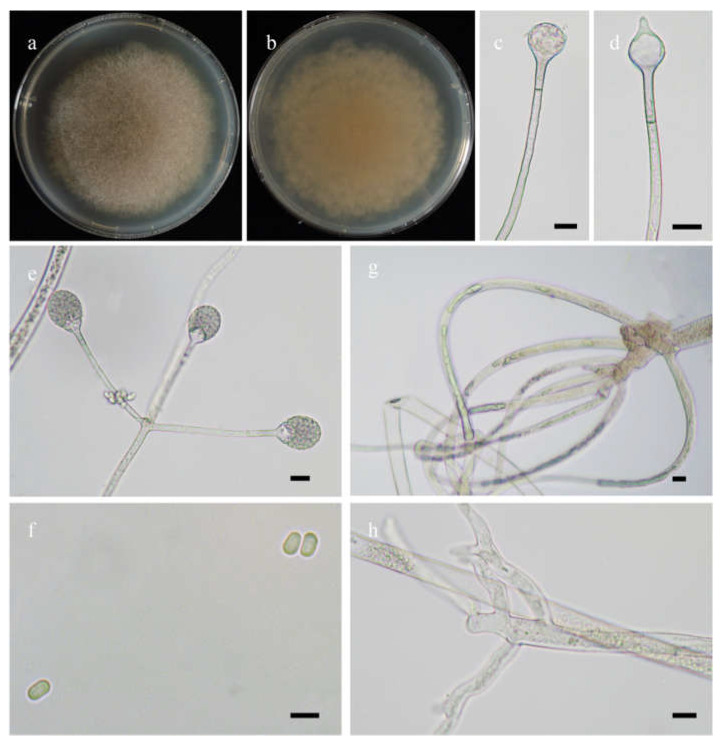
Morphologies of *Absidia sichuanensis* ex-holotype CGMCC 3.16258. (**a**,**b**). Colonies on MEA ((**a**), obverse, (**b**), reverse); (**c**), columellae without projections; (**d**), columellae with a projection; (**e**), sporangia in whorls; (**f**), sporangiospores; (**g**,**h**). Rhizoids—scale bars: (**c**–**e**,**g**,**h**), 10 μm, (**f**), 5 μm.

**Figure 7 jof-08-00471-f007:**
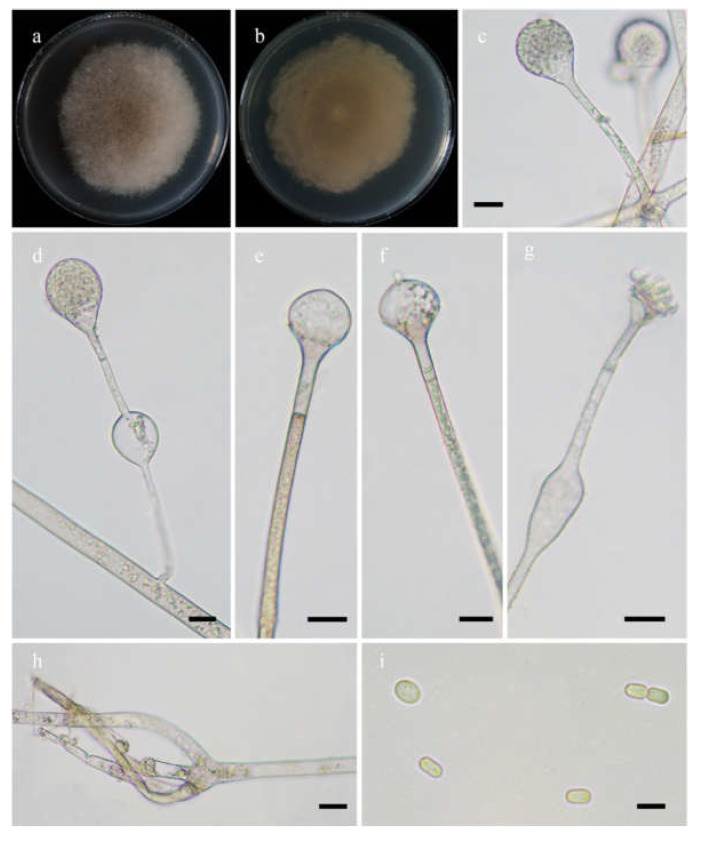
Morphologies of *Absidia yunnanensis* ex-holotype CGMCC 3.16259. (**a**,**b**). Colonies on MEA ((**a**), obverse, (**b**), reverse); (**c**), sporangia; (**d**), swellings below sporangia; (**e**–**g**), columellae; (**h**), rhizoids; (**i**), sporangiospores—scale bars: (**c**–**g**), 10 μm, (**h**), 20 μm, (**i**), 5 μm.

**Table 1 jof-08-00471-t001:** The origin of taxonomic types in *Absidia*.

Country	Type	Percentage (%)	Country	Type	Percentage (%)
China	9	19.6	Canada	1	2.2
Brazil	8	17.4	Cuba	1	2.2
USA	4	8.7	Egypt	1	2.2
France	3	6.5	Netherland	1	2.2
Korea	3	6.5	Mexico	1	2.2
Czechia	2	4.3	Pakistan	1	2.2
India	2	4.3	Switzerland	1	2.2
Thailand	2	4.3	Tanzania	1	2.2
Australia	1	2.2	Unknown	4	8.7

Note: These data are from the Index Fungorum (http://www.indexfungorum.org/, accessed on 1 March 2022) and [16].

**Table 2 jof-08-00471-t002:** Taxa information and GenBank accession numbers used in this study.

Species	Strain	GenBank Accession Nos.	
		**ITS**	**LSU rDNA**
** *Absidia abundans* **	**CGMCC 3.16255,** **T**	**ON074695**	**ON074683**
** *A. abundans* **	**XY09265**	**ON074697**	**ON074681**
** *A. abundans* **	**XY09274**	**ON074696**	**ON074682**
*A. aguabelensis*	URM 2813, T	MW763074	MW762874
*A. anomala*	CBS 125.68, T	NR_103626	NG_058562
*A. bonitoensis*	URM 7889, T	MN977786	MN977805
*A. caatingaensis*	URM 7156, T	KT308169	KT308171
*A. caerulea*	CBS 101.36	MH855718	MH867230
*A. caerulea*	FSU 767	AY944870	AF113443
*A. californica*	CBS 314.78	MH861141	MH872902
*A. californica*	FSU 4747	AY944872	EU736300
*A. californica*	FSU 4748	AY944873	EU736301
*A. cornuta*	URM 6100	MN625256	MN625255
*A. cuneospora*	CBS 101.59, T	NR_159602	NG058559
*A. cylindrospora*	CBS 100.08	JN205822	JN206588
*A. edaphica*	MFLU 20-0416	MT396372	MT393987
*A. frigida*	CGMCC 3.16201, T	OM108487	OM030223
*A. fusca*	CBS 102.35	NR103625	NG058552
*A. gemella*	CGMCC 3.16202, T	OM108488	OM030224
*A. glauca*	CBS 129233	MH865253	MH876693
*A. glauca*	CBS 101.08, T	NR_111658	NG_058550
*A. glauca*	FSU 660	AY944879	EU736302
*A. globospora*	CGMCC 3.16031, T	MW671537	MW671544
*A. globospora*	CGMCC 3.16035	MW671538	MW671545
*A. globospora*	CGMCC 3.16036	MW671539	MW671546
*A. healeyae*	UoMAU1, T	−	MT436027
*A. heterospora*	SHTH021	JN942683	JN982936
*A. idahoensis*	CBS 103.91, T	−	NG_058545
*A. inflata*	NRRL 6591	−	YES
*A. jindoensis*	CNUFC-PTI1-1, T	MF926622	MF926616
*A. koreana*	EML-IFS45-1, T	KR030062	KR030056
** *A. lobata* **	**CGMCC 3.16256** **, T**	**ON074690**	**ON074679**
** *A. lobata* **	**XY08832-1**	**ON074691**	**ON074680**
*A. longissima*	CGMCC 3.16203, T	OM108489	OM030225
*A. macrospora*	FSU 4746	AY944882	EU736303
*A. medulla*	CGMCC 3.16034, T	MW671542	MW671549
*A. montepascoalis*	URM 2818, T	NR_172995	−
*A. multispora*	URM 8210, T	MN953780	MN953782
*A. ovalispora*	CGMCC 3.16018, T	MW264071	MW264130
*A. panacisoli*	SYPF 7183, T	MF522181	MF522180
*A. pararepens*	CCF 6352	MT193669	MT192308
*A. pernambucoensis*	URM 7219, T	MN635568	MN635569
*A. pseudocylindrospora*	CBS 100.62, T	NR_145276	NG_058561
*A. pseudocylindrospora*	EML-FSDY6-2	KU923817	KU923814
*A. psychrophilia*	FSU 4745	AY944874	EU736306
** *A. radiata* **	**CGMCC 3.16257** **, T**	**ON074698**	**ON074684**
** *A. radiata* **	**XY09330-1**	**ON074699**	**ON074685**
*A. repens*	CBS 115583, T	NR103624	HM849706
*A. saloaensis*	URM 8209, T	MN953781	MN953783
** *A. sichuanensis* **	**CGMCC 3.16258** **, T**	**ON074692**	**ON074688**
** *A. sichuanensis* **	**XY09119**	**ON074693**	**−**
** *A. sichuanensis* **	**XY09633**	**ON074694**	**ON074689**
*A. soli*	MFLU 20-0414, T	MT396373	MT393988
*A. spinosa*	FSU 551	AY944887	EU736307
*A. stercoraria*	EML-DG8-1, T	NR_148090	KT921998
*A. terrestris*	FMR 14989	LT795003	LT795005
*A. turgida*	CGMCC 3.16032, T	MW671540	MW671547
** *A. yunnanensis* **	**CGMCC 3.16259** **, T**	**ON074700**	**ON074687**
** *A. yunnanensis* **	**XY09528**	**ON074701**	**ON074686**
*A. zonata*	CGMCC 3.16033, T	MW671541	MW671548
*Cunninghamella blakesleeana*	CBS 782.68	JN205869	MH870950
*C. elegans*	CBS 167.53	JN205882	HM849700

Notes: sequences obtained in this study are in bold. “T” after strain number represents strain type. “−” represents the absence of sequence in GenBank. “YES” represents the sequences from the whole-genome sequencing with BioProject accession PRJNA519280.

**Table 3 jof-08-00471-t003:** *Absidia* species distributions and ecological habitat in China.

Species	Location	Habitat	References
*A. abundans*	Yunnan	forest soil	This study
*A. cylindrospora*	Jilin, Taiwan, and Xinjiang	Soil and leaf litter	[27,50]
*A. frigida*	Xinjiang	soil	[16]
*A. gemella*	Xinjiang	soil	[16]
*A. glauca*	Beijing, Fujian, Hebei, Hubei, Inner Mongolia, Jilin, Liaoning, Shaanxi, Sichuan, Taiwan, Xinjiang, and Yunnan	soil, forest soil, air, rhizosphere soil of *Populus* and *Pinus*, and leaf litter	[27,50]
*A. globospora*	Hubei and Shaanxi	soil	[15]
*A. heterospora*	Guizhou, Sichuan, and Taiwan	soil	[27,50]
*A. idahoensis*	Yunnan	Soil and bee	[24]
*A. lobata*	Yunnan	rhizosphere soil of *Pinus yunnanensis*	This study
*A. longissima*	Yunnan	soil	[16]
*A. medulla*	Fujian, Jiangxi, and Yunnan	soil	[15]
*A. ovalispora*	Yunnan	soil	[14]
*A. panacisoli*	Yunnan	rhizosphere soil of *Panax notoginseng*	[16]
*A. pseudocylindrospora*	Taiwan	soil	[27,51]
*A. psychrophilia*	Heilongjiang and Taiwan	soil, leaf litter, rhizosphere soil of *Pinus*, and gland of *Ambrosia beetle*	[27,51]
*A. radiata*	Yunnan	forest soil	This study
*A. repens*	Xinjiang and Yunnan	soil, dung, paper, rhizosphere soil of *Ambrosia artemisiifolia*, and cave depositions	[27]
*A. sichuanensis*	Sichuan and Tibet	grassland soil, rhizosphere soil of *Picea asperata*, and rhizosphere soil of *Pinus yunnanensis*	This study
*A. spinosa*	Heilongjiang and Taiwan	soil, air, and leaf of *Comandra pallida*	[27,51]
*A. turgida*	Xinjiang	soil	[15]
*A. yunnanensis*	Yunnan	forest soil	This study
*A. zonata*	Beijing, Chongqing, Fujian, Shaanxi, and Yunnan	soil	[15]

**Table 4 jof-08-00471-t004:** Morphological features of the five new *Absidia* species and other six related species without DNA evidence.

Species	Colonies	Sporangiophore	Sporangia	Collars	Columellae	Projections	Sporangiospores	Zygospores	Temperature	References
** *Absidia abundans* **	on MEA at 27 °C for 7 days, 65 mm in diameter	unbranched, or simple if branched, monopodial, or sympodial	oval to subglobose, 8.0–16.5 × 8.5–16.0 µm	absent or present	subglobose or oval, 4.5–10.0 × 3.5–8.0 µm	single if present	cylindrical, oval or subglobose, 2.5–3.5 × 2.0–3.5 µm	unknown	mesophilic	This study
*A. clavata*	on MEA rapid growing	2–8 in whorls	pyriform to globose, 10.5–33.0 µm in diameter		globose to subglobose, 9.0–22.5 µm length	present, single	globose to oval, 2.0–5.0 × 1.8–3.2 µm	unknown	−	[53]
*A. dubia*	−	branched with a whorls	−	−	ovoid to nearly spatulate	−	oval, 2.2–4.2 × 2.2 µm	present	−	[54]
*A. egyptiaca*	−	−	usually globose, 25.3 µm	−	11.5–20.75 µm	−	ovoid to elliptical, 4.6–4.85 µm	heterozygotes, globose, up to 65 µm	−	MycoBank
*A. fassatiae*	on MEA for 6 days, 55 mm in diameter	3–9 in whorls	globose, 15–31 µm in diameter	−	hemispherical to coniform, 9–22 µm in diameter	−	cylindrical, 4.7–7.3 × 1.9–3.1 µm	unknown	−	[55]
** *A. lobata* **	on MEA at 20 °C for 7 days, 70 mm in diameter	in pairs, unbranched	pyriform, 22.0–43.5 × 18.5–31.0 µm	absent	subglobose to depressed globose, 12.0–26.5 × 11.0–25.0 µm	always present, single	mostly globose, occasionally subglobose, 2.5–3.0 µm in diameter	unknown	mesophilic	This study
*A. narayanae*	on SMA at 37 °C for 2 days, 75 mm in diameter	in group, up to 8	globose to subglobose,19.5–41.5 × 24.0–41.0 µm	absent or present	globose, hemispherical to mammiform, 14.5–34.0 × 12.0–34.0 µm	−	globose to ovate, 3.0–4.0 µm	unknown	Thermophilic or thermotolerant	[56]
** *A. radiata* **	on MEA at 27 °C for 7 days, 65 mm in diameter	1–5 in whorls, unbranched	pyriform or subglobose, 17.5–33.5 × 18.5–30.0 µm	absent	mostly oval, depressed globose, occasionally subglobose to globose, 13.5–22.5 × 14.0–24.0 µm	single if present	oval, 3.0–5.0 × 2.0–3.5 µm	unknown	mesophilic	This study
*A. reflexa*	−	singly from the stolons	pyriform	present	conical	one or several projections	spherical, 6 p. in diameter	unknown	−	[52]
** *A. sichuanensis* **	on MEA at 20 °C for 7 days, 75 mm in diameter	1–5 in whorls, unbranched	pyriform, 18.0–23.0 × 17.0–21.5 µm	absent or present	mostly subglobose, occasionally globose, 7.5–13.0 × 8.0–15.5 µm	single if present	cylindrical, 3.0–4.5× 2.0–2.5 µm	unknown	mesophilic	This study
** *A. yunnanensis* **	on MEA at 27 °C for 7 days, 65 mm in diameter	unbranched, monopodially branched, 2–5 in whorls	pyriform to subglobose, 16.0–27.5 × 16.5–27.0 µm	absent or present	oval, depressed globose, or occasionally globose, 6.5–18.5 × 7.5–22.0 µm	single if present	cylindrical, 3.5–5.0 × 2.0–4.0 µm	unknown	mesophilic	This study

Notes: New species proposed in this study are in bold. The “−” represents the absence of features.

## Data Availability

Sequences have been deposited in GenBank (Table 2).

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
