# Peer review of "Species Diversity and Ecological Habitat of Absidia (Cunninghamellaceae, Mucorales) with Emphasis on Five New Species from Forest and Grassland Soil in China"

_jof, 2022, doi:10.3390/jof8050471_

Round 1

Reviewer 1 Report

In this study, the authors describe five new Absidia species isolated from soil in China. Although every description of a new taxon is a valuable contribution to general knowledge of mycobiota, I’d recommend some significant modifications. The introduction needs more attention and should better introduce the research which was already done on Absidia genus. The methodological part should give more detailed information in order to enable their replication in the future. Below I’m presenting several detailed comments:

Line 28: The first Absidia was indeed described in 1876 and currently it is classified within Cunninghamellaceae but this family was not existing at the time. Several taxonomical changes were done to place Absidia within Cunninghamellaceae. Please reformulate to give precise information. It is also recommended, especially in the case of taxonomic papers, to refer to the original research placing Absidia within Cuninghamellaceae than just refer indexfungorum database.

Line 29-33: I would recommend developing this information. Although I agree that Gongronella was primarily described as Absidia it is a smaller taxonomic change than the division of the whole genus into two: Lichetheimia and Absidia.

Line 48: First, please correct properly the data taken from GBIF: https://www.gbif.org/citation-guidelines to acknowledge researchers who are sharing their biodiversity data through GBIF infrastructure. Second, please note that GBIF is not the only source of data on fungal distribution. Especially, in the taxonomic paper, I would recommend performing deeper literature and database research. For example, the data on Absidia distribution available here: https://globalfungi.com/ suggests a slightly different pattern. I can understand that Authors comment only on type specimens but I cannot understand why they are not commenting on other available materials.

Line 56: please avoid repetition of the term ‘obvious’.

Line 77: In the Introduction the Authors are citing data from GBIF. I would recommend (however it is not mandatory and should not affect the publication process) uploading and sharing their data on new taxa with GBIF too.

Lines 99-101: I miss the parameters of performed analysis. The algorithm used for alignment preparation is crucial for further analysis. The details like used algorithms and analysis parameters are more important than information on the program name. With sole information on the programs used no one would be able to repeat described analysis. Please develop this part.

Line 109: The analysis is based on 57 strains representing 41 species, which means that every species is represented by less than two sequences. The analysis of the speciation process and therefore of describing new taxa should rely on the analysis of gene flow between populations. A good proxy of such isolation is for example the measure of intraspecies and interspecies variation in sequences. While using only one sequence per species it is impossible to comment on intraspecies variability.

Line 144 (here and in all other species descriptions): ‘3.0–15.5’ are these measurements min and max sizes? What was the number of measurements performed (n)? Please indicate the mean and standard deviation. The range from 3 to 15 um is wide, so the information given in this form is not very informative.

Author Response

Thank you for your comments, and our reply is as follows:

In this study, the authors describe five new Absidia species isolated from soil in China. Although every description of a new taxon is a valuable contribution to general knowledge of mycobiota, I’d recommend some significant modifications. The introduction needs more attention and should better introduce the research which was already done on Absidia genus. The methodological part should give more detailed information in order to enable their replication in the future. Below I’m presenting several detailed comments:

Response: Thanks for your suggestion. In the section “Introduction”, we added more attention to introduce the research of the genus Absidia. And in the section “Materials and Methods”, we added more detailed information about Maximum Likelihood, Maximum Parsimony and Bayesian Inference analyses.

Line 28: The first Absidia was indeed described in 1876 and currently it is classified within Cunninghamellaceae but this family was not existing at the time. Several taxonomical changes were done to place Absidia within Cunninghamellaceae. Please reformulate to give precise information. It is also recommended, especially in the case of taxonomic papers, to refer to the original research placing Absidia within Cunninghamellaceae than just refer indexfungorum database.

Response: Thanks for your suggestion. We have added references to supported Absidia within Cunninghamellaceae. It says as “However, it was initially place in the family Absidiaceae [2]. With the development of molecular biology, it was grouped with Chlamydoabsidia Hesselt. & J.J. Ellis, Cunninghamella Matr., Gongronella Ribaldi, and Halteromyces Shipton & Schipper and Hesseltinella H.P. Upadhyay; and this group was nominated as the family Cunninghamellaceae [3-7].”

Line 29-33: I would recommend developing this information. Although I agree that Gongronella was primarily described as Absidia it is a smaller taxonomic change than the division of the whole genus into two: Lichetheimia and Absidia.

Response: Thanks for your suggestion. We have provided in the Introduction section with the following information “As research progressed, Absidias.l. has been well divided into three genera, i.e., Lichtheimia (thermotolerant, optimum growth temperature 37–45 °C), Absidia s.s. (mesophilic, optimum growth temperature 25–34 °C), and Lentamyces Kerst. Hoffm. & K. Voigt (parasitic on mucoralean fungi, optimum growth temperature 14–25 °C) [9-11].”

Line 48: First, please correct properly the data taken from GBIF: https://www.gbif.org/citation-guidelines to acknowledge researchers who are sharing their biodiversity data through GBIF infrastructure. Second, please note that GBIF is not the only source of data on fungal distribution. Especially, in the taxonomic paper, I would recommend performing deeper literature and database research. For example, the data on Absidia distribution available here: https://globalfungi.com/ suggests a slightly different pattern. I can understand that Authors comment only on type specimens but I cannot understand why they are not commenting on other available materials.

Response: Thanks. Firstly, we cited the GBIF database times according to the guideline, and they are references 21, 22 and 24. Secondly, we cited the GlobalFungi database as refernce 23. Thirdly, in Figure 1, we summarized the ecological habitat of type stains from Index Fungorum, GBIF and other studies, and species distribution of Absidia in China were concluded in the Table 3. Finally, previously studies were also cited as references 16, 25-33.

Line 56: please avoid repetition of the term ‘obvious’.

Response: Thanks. We followed your suggestion, and deleted ‘obvious’.

Line 77: In the Introduction the Authors are citing data from GBIF. I would recommend (however it is not mandatory and should not affect the publication process) uploading and sharing their data on new taxa with GBIF too.

Response: Thanks. We would like to share data by uploading on GBIF when the new taxa reported here are accepted.

Lines 99-101: I miss the parameters of performed analysis. The algorithm used for alignment preparation is crucial for further analysis. The details like used algorithms and analysis parameters are more important than information on the program name. With sole information on the programs used no one would be able to repeat described analysis. Please develop this part.

Response: Thanks for your suggestion. We developed this part as “Maximum Likelihood analysis was performed using the GTRGAMMA substitution model with 1 000 bootstrap replications. Maximum Parsimony analysis was conducted with 1 000 bootstrap replications using the heuristic search option with bisection and re-connection. Sequences were randomly added and max-trees were set to 5 000. For BI analysis, eight cold Markov chains run simultaneously for two million generations with the GTR + I + G model, sampling every 1 000 generations and with the first 25 % sampled tree being removed as burn-in.”

Line 109: The analysis is based on 57 strains representing 41 species, which means that every species is represented by less than two sequences. The analysis of the speciation process and therefore of describing new taxa should rely on the analysis of gene flow between populations. A good proxy of such isolation is for example the measure of intraspecies and interspecies variation in sequences. While using only one sequence per species it is impossible to comment on intraspecies variability.

Response: Thanks for your suggestion. In this study, we proposed five new species, each with two or three strains. In order to show the topology of the evolutionary history for the genus Absidia, we included one strain (usually the type strain) for most species. We added the BLAST top hits of their ITS sequences in the supplementary Table S1. In fact, for most other species, more ITS sequences identical to the one used were there in the GenBank database.

Line 144 (here and in all other species descriptions): ‘3.0–15.5’ are these measurements min and max sizes? What was the number of measurements performed (n)? Please indicate the mean and standard deviation. The range from 3 to 15 um is wide, so the information given in this form is not very informative.

Response: Thanks. In the section “2.2. Morphology and maximum growth temperature”, we detailed relative methods as “For morphological features, the minimum and maximum sizes based on a statistic of more than 50 measurements were adopted”. Traditionally, in the taxonomy of zygomycetes, morphological characteristics are described by using a range between the min and max sizes. We followed this tradition.

Reviewer 2 Report

Lines 16–17. Abstract. The first sentence makes no sense, as the clause beginning with “while” is a logical non-sequitur to the main sentence. Rewrite as “Although species of Absidia are known to be ubiquitous in soil, in animal dung, and in insect and plant debris, the species diversity of the genus and their ecological habitats have not been sufficiently investigated.”

Lines 18–19. Replace “soil” by “soils” and replace “and their novelty is supported by” by “with support provided by”.

Line 21. Replace “are mainly originated from soil” by “mainly occur in soil”.

Line 32. “Vuill” should read “Vuill.” In two places.

Line 35. Delete “are” after “sporangiophores”.

Line 36. Delete “are” after “rhizoids”.

Line 37. Delete “are” after “sporangia” and replace “bear” by “bearing”.

Line 38. Delete “are” after “zygospores”.

Line 39. Delete “are” after “temperatures”.

Line 41. Replace “letter” by “litter”.

Line 61. In Figure 1. Replace “Unknow” by “Unknown”.

Line 73. Absidia should be in italics.

Line 86. Replace “not growing” by “ceased growing”.

Line 104. Table 2. “A. caatinguensis” is presented as “A. caatingaensis” in Index Fungorum and should be spelled as such. Also, “coerulea” should be “caerulea”. These will also need to be corrected in Figure 2.

Lines 129–130. Replace “and a physiological trait of maximum growth temperatures” by “and the maximum growth temperature, a physiological trait, is also presented”.

Line 138. Replace “refer” by “refers”. Also, lines 168, 198, 226 and 257.

Line 284. Replace “described in the worldwide” by “described worldwide”.

Line 285. Replace “provided” by “provide”.

Lines 332–334. Replace these two sentences by “All species were found in soil, such as forest soil and rhizospheric soil, whereas other habitats, including leaf litter, dung, insect remains and plant leaves, recorded one to several species.”

Line 355. Replace “differ” by “differs”.

Line 365. This sentence needs revision, as α-galactosidase, laccase, chitosan and fatty acids are not industries. Please rewrite it.

Lines 367–368. Replace “were described” by “have previously been described”.

Line 372. Replace “like cases in” by “as in”.

Line 373. Replace “in the soil” by “in soil”.

Line 381. Replace “in Brazil” by “from Brazil”.

Line 383. Replace “area” by “areas” and replace “in forest soil” by “from forest soil”.

Author Response

Thank you for your comments, and our reply is as follows:

Lines 16–17. Abstract. The first sentence makes no sense, as the clause beginning with “while” is a logical non-sequitur to the main sentence. Rewrite as “Although species of Absidia are known to be ubiquitous in soil, in animal dung, and in insect and plant debris, the species diversity of the genus and their ecological habitats have not been sufficiently investigated.”

Response: Thank you. We accepted your suggestion.

Lines 18–19. Replace “soil” by “soils” and replace “and their novelty is supported by” by “with support provided by”.

Response: Thank you. We accepted your suggestion.

Line 21. Replace “are mainly originated from soil” by “mainly occur in soil”.

Response: Thank you. We corrected it as your suggestion.

Line 32. “Vuill” should read “Vuill.” In two places.

Response: Thank you. We have corrected this.

Line 35. Delete “are” after “sporangiophores”.

Response: Thank you. We have corrected this.

Line 36. Delete “are” after “rhizoids”.

Response: Thank you. We have corrected this.

Line 37. Delete “are” after “sporangia” and replace “bear” by “bearing”.

Response: Thank you. We have corrected this.

Line 38. Delete “are” after “zygospores”.

Response: Thank you. We have corrected this.

Line 39. Delete “are” after “temperatures”.

Response: Thank you. We have corrected this.

Line 41. Replace “letter” by “litter”.

Response: Thank you. We have corrected this.

Line 61. In Figure 1. Replace “Unknow” by “Unknown”.

Response: Thank you. We have corrected this.

Line 73. Absidia should be in italics.

Response: Thank you. We have corrected this.

Line 86. Replace “not growing” by “ceased growing”.

Response: Thanks. We corrected as “until the colonies stopped growing” in the revised manuscript.

Line 104. Table 2. “A. caatinguensis” is presented as “A. caatingaensis” in Index Fungorum and should be spelled as such. Also, “coerulea” should be “caerulea”. These will also need to be corrected in Figure 2.

Response: Thank you. The species names are presented in the Table 2 and Figure 2 based on the Index Fungorum.

Lines 129–130. Replace “and a physiological trait of maximum growth temperatures” by “and the maximum growth temperature, a physiological trait, is also presented”.

Response: Thank you. We accepted it.

Line 138. Replace “refer” by “refers”. Also, lines 168, 198, 226 and 257.

Response: Thank you. We have corrected these.

Line 284. Replace “described in the worldwide” by “described worldwide”.

Response: Thank you. We have corrected this.

Line 285. Replace “provided” by “provide”.

Response: Thank you. We corrected this.

Lines 332–334. Replace these two sentences by “All species were found in soil, such as forest soil and rhizospheric soil, whereas other habitats, including leaf litter, dung, insect remains and plant leaves, recorded one to several species.”

Response: Thank you very much. We replaced as your suggestion.

Line 355. Replace “differ” by “differs”.

Response: Thanks. We corrected this.

Line 365. This sentence needs revision, as α-galactosidase, laccase, chitosan and fatty acids are not industries. Please rewrite it.

Response: Thanks. We rewrite this sentence. “Species of Absidia synthesized important metabolites, such as α-galactosidase, lac-case, chitosan and fatty acids [58-60], and our results provide a basis for their applications.”

Lines 367–368. Replace “were described” by “have previously been described”.

Response: Thank you. We corrected this.

Line 372. Replace “like cases in” by “as in”.

Response: Thank you very much.

Line 373. Replace “in the soil” by “in soil”.

Response: Thank you.

Line 381. Replace “in Brazil” by “from Brazil”.

Response: Thank you.

Line 383. Replace “area” by “areas” and replace “in forest soil” by “from forest soil”.

Response: Thanks.

Reviewer 3 Report

Species diversity and ecological habitat of Absidia (Cunning-hamellaceae, Mucorales) with emphasis on five new species from forest and grassland soil in China

The author propose 5 new species of Absidia. The manuscript is well organized with minimal grammatical errors however the manuscript needs additional information to be published.

Primary Concern: The author completed phylogenetic analyses on a total of 35 species out of the 46/47 current species. This is likely because there is no molecular data for the remaining species. The authors must demonstrate that any existing species without DNA evidence is different from the ones they are proposing based on morphology or other criteria. This is a requirement to publish new species.

Related Concern: There is some discrepancy with the number of type species in Table 1. I believe the type species in table 1 adds to 47, while the authors state only 46 which is minor but needs to be clarified because in the end these may make 52, not 51 total species. Please review and adjust accordingly.

Other minor Line items:

Line 21: … which are mainly originated from soil.

Please consider revising awkward as written.

Line 33: … consider revising to: As research progressed, …

Line 40: Please review the total number of species as Table 1 has 47.

Line 43: Please consider revising to: …Absidia are mainly isolated from soil [11,12].

Line 48 Please spell out Absidia when it’s the first word in a sentence.

Table 1: Please double check the numbers.

Line 70: Please remove PDA: as it is the only time it is abbreviated in the manuscript.

Line 73: Please italicize Absidia

Line 86: Please consider revising to: …until the colonies stopped growing.

Line 97: Please consider revising to: … Sanger sequencing of PCR products…

Line 111: Spell out abbreviations if they are the first word in a sentence: please spell out MP.

Line 114: Please consider revising to: model for Baysian Inference (BI) was GTR…

Line 115: Please consider revising to: split frequenc was less than 0.01.

Line 154: Please review as I am unclear how the following can occur: “Projections rarely absent, always present, single if present.” How can they be rarely absent but always present?

Line 283: Depending on Table 1, the total may be 52.

Line 284: Please consider revising to: Asidia has been described worldwide.

Line 333-334: The subject “other habitat” and verb “were” do not agree. Consider revising.

Author Response

Thank you for your comments, and our reply is as follows:

Species diversity and ecological habitat of Absidia (Cunninghamellaceae, Mucorales) with emphasis on five new species from forest and grassland soil in China

The author proposed 5 new species of Absidia. The manuscript is well organized with minimal grammatical errors however the manuscript needs additional information to be published.

Primary Concern: The author completed phylogenetic analyses on a total of 35 species out of the 46/47 current species. This is likely because there is no molecular data for the remaining species. The authors must demonstrate that any existing species without DNA evidence is different from the ones they are proposing based on morphology or other criteria. This is a requirement to publish new species.

Response: Thanks for your suggestion. In this study, we added DNA evidence for five more species, and the remaining six species A. clavata, A. dubia, A. egyptiaca, A. fassatiae, A. narayanae, and A. reflex, are still without DNA sequences. A morphological feature table of these six species is listed in the revised manuscript, compared with those of the five new species proposed in this study (Table 4).

Related Concern: There is some discrepancy with the number of type species in Table 1. I believe the type species in table 1 adds to 47, while the authors state only 46 which is minor but needs to be clarified because in the end these may make 52, not 51 total species. Please review and adjust accordingly.

Response: Thank you. In the Table 1, type strains with an unknown country information are 4, instead of 5, and thus we corrected this mistake. Consequently, a totally 51 species was described, not 52.

Other minor Line items:

Line 21: … which are mainly originated from soil.

Please consider revising awkward as written.

Response: Thank you. We corrected this as “which mainly occur in soil”.

Line 33: … consider revising to: As research progressed, …

Response: Thank you. We have corrected as your suggestion.

Line 40: Please review the total number of species as Table 1 has 47.

Response: Thank you. The species number is 46, not 47. Because “Unknown” is 4, not 5. We have corrected.

Line 43: Please consider revising to: …Absidia are mainly isolated from soil [11,12].

Response: Thank you. We have corrected as your suggestion.

Line 48 Please spell out Absidia when it’s the first word in a sentence.

Response: Thank you. We followed this rule throughout the manuscript.

Table 1: Please double check the numbers.

Response: Thanks, the numbers have been corrected.

Line 70: Please remove PDA: as it is the only time it is abbreviated in the manuscript.

Response: Thank you. The PDA abbreviation was removed in the revised manuscript.

Line 73: Please italicize Absidia

Response: Thanks. We italicized the genus name.

Line 86: Please consider revising to: …until the colonies stopped growing.

Response: Thanks. We corrected as “until the colonies stopped growing” in the revised manuscript.

Line 97: Please consider revising to: … Sanger sequencing of PCR products…

Response: Thanks. We accepted this suggestion.

Line 111: Spell out abbreviations if they are the first word in a sentence: please spell out MP.

Response: Thanks. “Maximum Parsimony (MP)” is adopted.

Line 114: Please consider revising to: model for Bayesian Inference (BI) was GTR…

Response: Thanks. We accepted this suggestion.

Line 115: Please consider revising to: split frequencies was less than 0.01.

Response: Thanks. We corrected as “split frequencies was 0.07357”.

Line 154: Please review as I am unclear how the following can occur: “Projections rarely absent, always present, single if present.” How can they be rarely absent but always present?

Response: Thanks. We corrected as “Projections single if present”.

Line 283: Depending on Table 1, the total may be 52.

Response: The mistake has been corrected.

Line 284: Please consider revising to: Absidia has been described worldwide.

Response: Thanks. We accepted this suggestion.

Line 333-334: The subject “other habitat” and verb “were” do not agree. Consider revising.

Response: Thanks. We replaced these two sentences by “All species were found in soil, such as forest soil and rhizospheric soil, whereas other habitats, including leaf litter, dung, insect remains and plant leaves, recorded one to several species.” according to other reviewer’s suggestion.

Reviewer 4 Report

Comment 1: You have recently published paper describing three new species of Absidia. Why you did not cite it ?  Did you isolate them together with the five new species of the current study?

Comment 2: Page 3 line 69 Add the date of collection.

Comment 3: Add the BLAST top hist of ITS sequences in the supplementary information .

Comment 4: Table 1 add reference

Comment 5: Page 3 Line 73  Absidia in italics

Comment 6: Page 3 Line 86 until the colonies stop growing instead of until the colonies not growing

Comment 7: Key to the species of Absidia in China:  By adding  the number of species having the same characteristic does not provide key to the reader. I would recommend  to cite  Absidia species in a table and mark the respective characteristic (a separate column for each characteristic).  

Comment 8: Table 3 add references

Author Response

Thank you for your comments, and our reply is as follows:

Comment 1: You have recently published paper describing three new species of Absidia. Why you did not cite it?  Did you isolate them together with the five new species of the current study?

Response: Yes, we have recently published a paper describing three new species of Absidia. In this paper, the three species are cited, containing the sequences, phylogeny analyses and updated the key.

Comment 2: Page 3 line 69 Add the date of collection.

Response: Thanks. The date of collection is placed in the “Isolation and strains”.

Comment 3: Add the BLAST top hist of ITS sequences in the supplementary information.

Response: Thanks. The BLAST top hit of ITS sequences was added in Supplementary Table S1.

Comment 4: Table 1 add reference.

Response: Thanks. The references were added.

Comment 5: Page 3 Line 73 Absidia in italics.

Response: Thanks. We have italicized it.

Comment 6: Page 3 Line 86 until the colonies stop growing instead of until the colonies not growing.

Response: Thanks. We corrected as “colonies stopped growing” in the revised manuscript.

Comment 7: Key to the species of Absidia in China: By adding the number of species having the same characteristic does not provide key to the reader. I would recommend to cite Absidia species in a table and mark the respective characteristic (a separate column for each characteristic).

Response: Thanks for your suggestion. A characteristic table of the five new species was added in Table 4, in comparison with the species without DNA data.

Comment 8: Table 3 add references

Response: Thanks. The references were added.

Round 2

Reviewer 1 Report

This is revised version of the manuscript that I've already reviewed. The Authors followed all recommendation and the manuscript is significantly improved. 

Reviewer 3 Report

Thank you for adding the remaining species and providing the morphological comparison table.